# Importance Weighted Hierarchical Variational Inference

**Artem Sobolev**
Samsung AI Center Moscow, Russia
asobolev@bayesgroup.ru

**Dmitry Vetrov**
Samsung AI Center Moscow, Russia
NRU HSE,* Moscow, Russia

## Abstract

Variational Inference is a powerful tool in the Bayesian modeling toolkit, however, its effectiveness is determined by the expressivity of the utilized variational distributions in terms of their ability to match the true posterior distribution. In turn, the expressivity of the variational family is largely limited by the requirement of having a tractable density function. To overcome this roadblock, we introduce a new family of variational upper bounds on a log marginal density in the case of hierarchical models (also known as latent variable models). We then derive a family of increasingly tighter variational lower bounds on the otherwise intractable standard evidence lower bound for hierarchical variational distributions, enabling the use of more expressive approximate posteriors. We show that previously known methods, such as Hierarchical Variational Models, Semi-Implicit Variational Inference and Doubly Semi-Implicit Variational Inference can be seen as special cases of the proposed approach, and empirically demonstrate superior performance of the proposed method in a set of experiments.

## 1 Introduction

Bayesian Inference is an important statistical tool. However, exact inference is possible only in a small class of conjugate problems, and for many practically interesting cases, one has to resort to Approximate Inference techniques. Variational Inference (Hinton and van Camp, 1993; Waterhouse et al., 1996; Wainwright et al., 2008) being one of them is an efficient and scalable approach that gained a lot of interest in recent years due to advances in Neural Networks.

However, the efficiency and accuracy of Variational Inference heavily depend on how close an approximate posterior is to the true posterior. As a result, Neural Networks' universal approximation abilities and great empirical success propelled a lot of interest in employing them as powerful sample generators that are trained to output samples from approximate posterior when fed some standard noise as input (Nowozin et al., 2016; Goodfellow et al., 2014; MacKay, 1995). Unfortunately, a significant obstacle in this direction is a need for a tractable density $q(z \mid x)$, which in general requires intractable integration. A theoretically sound approach then is to give tight lower bounds on the intractable term – the differential entropy of $q(z|x)$, which is easy to recover from upper bounds on the log marginal density. One such bound was introduced by Agakov and Barber (2004); however it's tightness depends on the auxiliary variational distribution. Yin and Zhou (2018) suggested a multisample surrogate whose quality is controlled by the number of samples.

In this paper we consider hierarchical variational models (Ranganath et al., 2016; Salimans et al., 2015; Agakov and Barber, 2004) where the approximate posterior $q(z \mid x)$ is represented as a mixture of tractable distributions $q(z|\psi, x)$ over some tractable mixing distribution $q(\psi|x)$: $q(z|x) = \int q(z|\psi, x)q(\psi|x)d\psi$. We show that such variational models contain *semi-implicit models* where

$q(\psi|x)$ is allowed to be arbitrarily complicated while being reparametrizable (Yin and Zhou, 2018). To overcome the need for the closed-form marginal density $q(z|x)$ we then propose a novel family of tighter bounds on the log marginal likelihood $\log p(x)$, which can be shown to generalize many previously known bounds: Hierarchical Variational Models (Ranganath et al., 2016) also known as auxiliary VAE bound (Maaløe et al., 2016), Semi-Implicit Variational Inference (Yin and Zhou, 2018) and Doubly Semi-Implicit Variational Inference (Molchanov et al., 2018). At the core of our work lies a novel variational upper bound on the log marginal density, which we apply to the evidence lower bound (ELBO) to enable hierarchical approximate posteriors. Finally, our method can be combined with the multisample bound of Burda et al. (2015) to tighten the log marginal likelihood lower bound even further.

## 2 Background

Having a hierarchical model $p_\theta(x) = \int p_\theta(x \mid z)p_\theta(z)dz$ for observable objects $x$, we are interested in two tasks: inference and learning. The problem of Bayesian inference is that of finding the true posterior distribution $p_\theta(z \mid x)$, which is often intractable and thus is approximated by some $q_\phi(z \mid x)$. The problem of learning is that of finding parameters $\theta$ s.t. the marginal model distribution $p_\theta(x)$ approximates the true data-generating process of $x$ as good as possible, typically in terms of KL-divergence, which corresponds to the Maximum Likelihood Estimation problem.

Variational Inference provides a way to solve both tasks simultaneously by lower-bounding the intractable log marginal likelihood $\log p_\theta(x)$ with the Evidence Lower Bound (ELBO) using a posterior approximation $q_\phi(z \mid x)$:

$$\log p_\theta(x) \geq \log p_\theta(x) - D_{KL}(q_\phi(z \mid x) \mid\mid p_\theta(z|x)) = \mathop{\mathbb{E}}_{q_\phi(z|x)} \log \frac{p_\theta(x, z)}{q_\phi(z \mid x)}$$

The bound requires analytically tractable densities for both $q_\phi(z \mid x)$ and $p_\theta(x, z)$. Since the ELBO is a biased objective, maximizing it w.r.t. $\theta$ not only (or necessarily) maximizes the log marginal likelihood, but also minimizes the KL divergence, constraining the true posterior $p_\theta(z|x)$ to stay close to the approximate one $q_\phi(z|x)$ and thus limiting the expressivity of $p_\theta(x)$. Such *variational bias* can be reduced by tightening the bound. In particular, Burda et al. (2015) proposed a family of tighter multisample bounds, generalizing the ELBO. We call it the IWAE bound:

$$\log p_\theta(x) \geq \mathop{\mathbb{E}}_{q_\phi(z_{1:M}|x)} \log \frac{1}{M} \sum_{m=1}^{M} \frac{p_\theta(x, z_m)}{q_\phi(z_m \mid x)},$$

where from now on we write $q_\phi(z_{1:M}|x) = \prod_{m=1}^{M} q_\phi(z_m|x)$ for brevity. This bound has been shown (Domke and Sheldon, 2018) to be a tractable lower bound on ELBO for a variational distribution that has been obtained by *self-normalizing importance sampling*, or a special case of Sequential Monte Carlo (Maddison et al., 2017; Naesseth et al., 2017; Le et al., 2017). However, the price of this increased tightness is higher computation complexity that mostly stems from increased number of evaluations of high-dimensional decoder $p_\theta(x|z)$. We thus focus on learning more expressive posterior approximations to be used in the ELBO – a special case of $M = 1$.

In the direction of improving the single-sample ELBO Agakov and Barber (2004); Salimans et al. (2015); Maaløe et al. (2016); Ranganath et al. (2016) proposed to use a hierarchical variational model (HVM) for $q_\phi(z \mid x) = \int q_\phi(z \mid x, \psi)q_\phi(\psi \mid x)d\psi$ with explicit joint density $q_\phi(z, \psi \mid x)$, where $\psi$ are *auxiliary latent variables*. Since the standard ELBO is now intractable due to the $\log q_\phi(z \mid x)$ term, the following variational lower bound on the ELBO is proposed. The tightness of this bound is controlled by the *auxiliary variational distribution* $\tau_\eta(\psi \mid x, z)$:

$$\log p_\theta(x) \geq \mathop{\mathbb{E}}_{q_\phi(z|x)} \log \frac{p_\theta(x, z)}{q_\phi(z \mid x)} \geq \mathop{\mathbb{E}}_{q_\phi(z, \psi|x)} \left[ \log p_\theta(x, z) - \log \frac{q_\phi(z, \psi \mid x)}{\tau_\eta(\psi \mid x, z)} \right] \qquad (1)$$

However, this bound now introduces *auxiliary variational bias*: the gap to the true ELBO is equal to $D_{KL}(q_\phi(\psi|z, x) \mid\mid \tau_\eta(\psi|z, x))$, which prevents learning expressive $q_\phi(z|x)$.

Recently Yin and Zhou (2018) introduced *semi-implicit models*: hierarchical models $q_\phi(z \mid x) = \int q_\phi(z \mid \psi, x)q_\phi(\psi \mid x)d\psi$ with implicit but reparametrizable $q_\phi(\psi \mid x)$ and explicit $q_\phi(z \mid \psi, x)$,

and suggested the following surrogate objective, which was later shown to be a lower bound (the SIVI bound) for all finite $K$ by Molchanov et al. (2018):

$$\log p_\theta(x) \geq \mathop{\mathbb{E}}_{q_\phi(z, \psi_0 | x)} \mathop{\mathbb{E}}_{q_\phi(\psi_{1:K} | x)} \log \frac{p_\theta(x, z)}{\frac{1}{K+1} \sum_{k=0}^{K} q_\phi(z | \psi_k, x)} \tag{2}$$

An appealing property of this bound is that it gets tighter as the number of samples $K$ increases and unlike the IWAE bound, it performs multiple computations in the smaller latent space. That said, SIVI can be generalized to use multiple samples $z$ similar to the IWAE bound (Burda et al., 2015) in an efficient way by reusing samples $\psi_{1:K}$ for different $z_m$:

$$\log p(x) \geq \mathbb{E} \left[ \log \frac{1}{M} \sum_{m=1}^{M} \frac{p_\theta(x, z_m)}{\frac{1}{K+1} \sum_{k=0}^{K} q_\phi(z_m | x, \psi_{m,k})} \right] \tag{3}$$

Where the expectation is taken over $q_\phi(\psi_{1:M,0}, z_{1:M} | x)$ and $\psi_{m,1:K} = \psi_{1:K} \sim q_\phi(\psi_{1:K} | x)$ is the same set of $K$ i.i.d. random variables for all $m^2$. Importantly, this estimator has $O(M + K)$ sampling complexity for $\psi$, unlike the naive approach, leading to $O(MK + M)$ sampling complexity. We will get back to this discussion in section 4.1.

## 2.1 SIVI Insights

Here we outline SIVI's points of weaknesses and identify certain traits that make it possible to generalize the method and bridge it with the prior work.

First, note that both SIVI bounds (2) and (3) use samples from $q_\phi(\psi_{1:K} | x)$ to describe $z$, and in high dimensions one might expect that such "uninformed" samples would miss most of the time, resulting in near-zero likelihood $q_\phi(z | \psi_k, x)$ and thus reduce the "effective sample size". Therefore it is expected that in higher dimensions it would take many samples to accurately cover the regions high probability of $q_\phi(\psi | z, x)$ for a given $z$. Instead, ideally, we would like to target such regions directly while keeping the lower bound guarantees.

Another important observation that we'll make use of is that many such semi-implicit models can be equivalently reformulated as a mixture of two explicit distributions: due to reparametrizability of $q_\phi(\psi | x)$ we have $\psi = g_\phi(\varepsilon | x)$ for some $\varepsilon \sim q(\varepsilon)$ with tractable density. We can then consider an equivalent hierarchical model $q_\phi(z|x) = \int q_\phi(z | g_\phi(\varepsilon | x), x) q(\varepsilon) d\varepsilon$ that first samples $\varepsilon$ from some simple distribution, transforms this sample $\varepsilon$ into $\psi$ and then generates samples from $q_\phi(z | \psi, x)$. Thus from now on we'll assume both $q_\phi(z | \psi, x)$ and $q_\phi(\psi | x)$ have a tractable density, yet $q_\phi(z | \psi, x)$ can depend on $\psi$ in an arbitrarily complex way, making analytic marginalization intractable.

## 3 Importance Weighted Hierarchical Variational Inference

Having intractable $\log q_\phi(z | x)$ as a source of our problems, we seek a tractable and efficient *upper* bound, which is provided by the following theorem:

**Theorem** (Log marginal density upper bound). *For any $q(z, \psi | x)$, $K \in \mathbb{N}_0$ and $\tau(\psi | z, x)$ (under mild regularity conditions) consider the following*

$$\mathcal{U}_K = \mathop{\mathbb{E}}_{q(\psi_0 | x, z)} \mathop{\mathbb{E}}_{\tau(\psi_{1:K} | z, x)} \log \left( \frac{1}{K+1} \sum_{k=0}^{K} \frac{q(z, \psi_k | x)}{\tau(\psi_k | z, x)} \right)$$

*Then the following holds:*

1. $\mathcal{U}_K \geq \log q(z | x)$

2. $\mathcal{U}_K \geq \mathcal{U}_{K+1}$

3. $\lim_{K \to \infty} \mathcal{U}_K = \log q(z | x)$

*Proof.* See Appendix for Theorem C.1. □

The proposed upper bound provides a variational alternative to MCMC-based upper bounds (Grosse et al., 2015) and complements the standard Importance Weighted stochastic lower bound of Burda et al. (2015) on the log marginal density:

$$\mathcal{L}_K = \mathop{\mathbb{E}}_{\tau(\psi_{1:K}|z,x)} \log \left( \frac{1}{K} \sum_{k=1}^{K} \frac{q(z, \psi_k \mid x)}{\tau(\psi_k \mid z, x)} \right) \leq \log q(z \mid x)$$

### 3.1 Tractable lower bounds on log marginal likelihood with a hierarchical proposal

The proposed upper bound $\mathcal{U}_K$ allows us to lower bound the otherwise intractable ELBO in case of hierarchical $q_\phi(z \mid x)$, leading to **Importance Weighted Hierarchical Variational Inference (IWHVI)** lower bound:

$$\log p_\theta(x) \geq \mathop{\mathbb{E}}_{q_\phi(z|x)} \log \frac{p_\theta(x,z)}{q_\phi(z \mid x)} \geq \mathop{\mathbb{E}}_{q_\phi(z,\psi_0|x)} \mathop{\mathbb{E}}_{\tau_\eta(\psi_{1:K}|z,x)} \log \frac{p_\theta(x,z)}{\frac{1}{K+1} \sum_{k=0}^{K} \frac{q_\phi(z,\psi_k|x)}{\tau_\eta(\psi_k|z,x)}} \tag{4}$$

Crucially, we merged expectations over $q_\phi(z|x)$ and $q_\phi(\psi_0|x,z)$ into an expectation over the joint distribution $q_\phi(\psi_0, z|x)$, which admits a more favorable factorization into $q_\phi(\psi_0|x)q_\phi(z|x,\psi_0)$, and samples from the later are easy to simulate for the Monte Carlo-based estimation.

IWHVI introduces an additional auxiliary variational distribution $\tau_\eta(\psi \mid x, z)$ that is learned by maximizing the bound w.r.t. its parameters $\eta$. While the optimal distribution is[3] $\tau(\psi \mid z, x) = q(\psi \mid z, x)$, one can see that some particular choices of auxiliary distributions and number of samples render previously known methods like DSIVI, SIVI and HVM as special cases (see appendix A). Since the bound (4) can be seen as variational generalization of SIVI (2) or as multisample generalization of HVM (1), it has capacity to give tighter bound on the log marginal likelihood and reduce the auxiliary variational bias, which should lead to more expressive variational approximations $q_\phi(z|x)$ and reduce in the variational bias.

One could also consider a hierarchical prior $p(z)$ (Atanov et al., 2019) and give tight multisample variational lower bounds using the bound $\mathcal{L}_K$ of Burda et al. (2015). Such nested variational inference is out of the scope of this paper, so we leave this direction to future work. Notably, the combination of two bounds can give multisample variational sandwich bounds on the KL divergence between hierarchical models (See appendix B).

## 4 Multisample Extensions

Multisample bounds similar to the proposed one have already been studied extensively. In this section, we relate our results to such prior work.

### 4.1 Multisample Bound and Complexity

One can generalize the bound (4) further in a way similar to the IWAE multisample bound (Burda et al., 2015), leading to the **Doubly Importance Weighted Hierarchical Variational Inference (DIWHVI)** (see Theorem C.4):

$$\log p_\theta(x) \geq \mathbb{E} \left[ \log \frac{1}{M} \sum_{m=1}^{M} \frac{p_\theta(x, z_m)}{\frac{1}{K+1} \sum_{k=0}^{K} \frac{q_\phi(z_m, \psi_{m,k}|x)}{\tau_\eta(\psi_{m,k}|z_m,x)}} \right] \tag{5}$$

Where the expectation is taken over the same generative process as in eq. (4), independently repeated $M$ times:

1. Sample $\psi_{m,0} \sim q_\phi(\psi \mid x)$ for $1 \leq m \leq M$
2. Sample $z_m \sim q_\phi(z \mid x_n, \psi_{m,0})$ for $1 \leq m \leq M$

3. Sample $\psi_{m,k} \sim \tau_\eta(\psi \mid z_m, x)$ for $1 \le m \le M$ and $1 \le k \le K$

The price of the tighter bound (5) is quadratic sample complexity as it requires $M(1 + K)$ samples of $\psi$. Unfortunately, the DIWHVI cannot benefit from the sample reuse trick of the SIVI that leads to the bound (3). The reason for this is that the bound (5) requires all terms in the outer denominator (the $\log q_\phi(z \mid x)$ estimate) to use the same distribution $\tau_\eta(\psi|x, z)$, whereas by its very nature it should be very different for different $z_m$. A viable option, though, is to consider a multisample-conditioned $\tau_\eta(\psi \mid z_{1:M})$ that is invariant to permutations of $z$. We leave a more detailed investigation to a future work.

Runtime-wise when compared to the multisample SIVI (3) the DIWHVI requires additional $O(M)$ passes to generate $\tau(\psi \mid x, z_m)$ distributions. However, since the SIVI requires a much larger number of samples $K$ to reach the same level of accuracy (see section 6.1) that are all then passed through a network to generate $q_\phi(z_m \mid x, \psi_{mk})$ distributions, the extra $\tau_\eta$ computation is likely to either bear a minor overhead, or be completely justified by reduced $K$. This is particularly true in the IWHVI case ($M = 1$) where IWHVI's single extra pass that generates $\tau_\eta(\psi|x, z)$ is dominated by $K + 1$ passes that generate $q_\phi(z|x, \psi_k)$.

## 4.2 Signal to Noise Ratio

Rainforth et al. (2018) have shown that multisample bounds (Burda et al., 2015; Nowozin, 2018) behave poorly during the training phase, having more noisy inference network's gradient estimates, which manifests itself in decreasing Signal-to-Noise Ratio (SNR) as the number of samples increases. This raises a natural concern whether the same happens in the proposed model as $K$ increases. Tucker et al. (2019) have shown that upon a careful examination a REINFORCE-like (Williams, 1992) term can be seen in the gradient estimate, and REINFORCE is known for its typically high variance (Rezende et al., 2014). Authors further suggest to apply the reparametrization trick (Kingma and Welling, 2013) the second time to obtain a reparametrization-based gradient estimate, which is then shown to solve the decreasing SNR problem. The same reasoning can be applied to our bound, and we provide further details and experiments in appendix D, developing an IWHVI-DReG gradient estimator. We conclude that the problem of decreasing SNR exists in our bound as well, and is mitigated by the proposed gradient estimator.

## 4.3 Debiasing the bound

Nowozin (2018) has shown that the standard IWAE can be seen as a biased estimate of the log marginal likelihood with the bias of order $O(1/M)$. They then suggested to use Generalized Jackknife of $d$-th order to reuse these $M$ samples and come up with an estimator with a smaller bias of order $O(1/M^d)$ at the cost of higher variance and losing lower bound guarantees. Again, the same idea can be applied to our estimate; we leave further details to appendix E. We conclude that this way one can obtain better estimates of the log marginal density, however since there is no guarantee that the obtained estimator gives an upper or a lower bound, we chose not to use it in experiments.

# 5 Related Work

More expressive variational distributions have been under an active investigation for a while. While we have focused our attention to approaches employing hierarchical models via bounds, there are many other approaches, roughly falling into two broad classes.

One possible approach is to augment some standard $q(z|x)$ with help of copulas (Tran et al., 2015), mixtures (Guo et al., 2016; Gershman et al., 2012), or invertible transformations with tractable Jacobians also known as normalizing flows (Rezende and Mohamed, 2015; Kingma et al., 2016; Dinh et al., 2016; Papamakarios et al., 2017), all while preserving the tractability of the density. Kingma and Dhariwal (2018) have demonstrated that flow-based models are able to approximate complex high-dimensional distributions of real images, but the requirement for invertibility might lead to inefficiency in parameters usage and does not allow for abstraction as one needs to preserve dimensions.

An alternative direction is to embrace implicit distributions that one can only sample from, and overcome the need for tractable density using bounds or estimates (Huszár, 2017). Methods based on

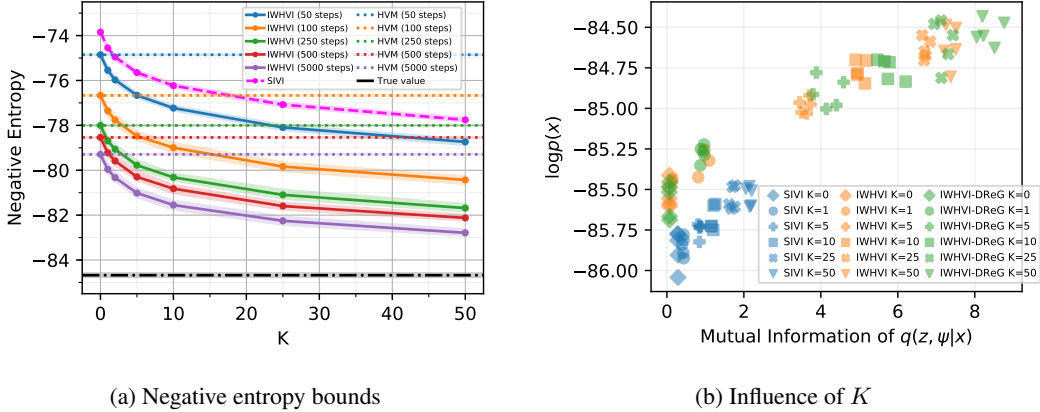

(a) Negative entropy bounds        (b) Influence of $K$

Figure 1: (*a*) Negative entropy upper bounds for 50-dimensional Laplace distribution. Shaded area denotes 90% confidence interval computed over 50 independent runs for each $K$. (*b*) Final log marginal likelihood $\log p(x)$ estimates and expected $\mathrm{MI}[q(z, \psi|x)]$ for IWHVI or SIVI-based VAEs trained with different $K$. Each model was trained and plotted 5 times.

estimates (Mescheder et al., 2017; Shi et al., 2017; Tran et al., 2017), for example, via the Density Ratio Estimation trick (Goodfellow et al., 2014; Uehara et al., 2016; Mohamed and Lakshminarayanan, 2016), typically estimate the densities indirectly utilizing an auxiliary critic and hide dependency on variational parameters $\phi$, hence biasing the optimization procedure. Major disadvantage of such methods is that they lose lower bound guarantees.

Titsias and Ruiz (2018) have shown that in the gradient-based ELBO optimization in case of a hierarchical model with tractable $q_\phi(z \mid \psi)$ and $q_\phi(\psi)$ one does not need the log marginal density $\log q_\phi(z \mid x)$ per se, only its gradient, which can be estimated using MCMC. Although unbiased, the MCMC-based posterior sampling has sequential nature (one needs to to perform chain burn-in to decorrelate $\psi'$ from its initial value) not amendable to efficient parallelization available with modern hardware such as GPUs, which complicates scaling the method to larger problems. In contrast, our method allows parallel computation of $K$ density ratios in $\mathcal{U}_K$.

The core contribution of the paper is a novel upper bound on log marginal density. Previously, Dieng et al. (2017); Kuleshov and Ermon (2017) suggested using $\chi^2$-divergence to give a variational upper bound to the log marginal density. However, their bound was not an expectation of a random variable, but instead a logarithm of the expectation, preventing unbiased stochastic optimization. Jebara and Pentland (2001) reverse Jensen's inequality to give a variational upper bound in case of mixtures of exponential family distributions by extensive use of the problem's structure. Related to our core idea of joint sampling $z$ and $\psi_0$ in (7) is an observation of Grosse et al. (2015) that Annealed Importance Sampling (AIS, Neal (2001)) ran backward from the auxiliary variable sample $\psi_0$ gives an unbiased estimate of $1/q(z \mid x)$, and thus can also be used to upper bound the log marginal density. However, AIS-based estimation is too computationally expensive to be used during training.

# 6 Experiments

## 6.1 Toy Experiment

As a toy experiment we consider a 50-dimensional factorized standard Laplace distribution $q(z)$ as a hierarchical scale-mixture model:

$$q(z) = \prod_{d=1}^{50} \mathrm{Laplace}(z_d \mid 0, 1) = \int \prod_{d=1}^{50} \mathcal{N}(z_d \mid 0, \psi_d) \mathrm{Exp}(\psi_d \mid \tfrac{1}{2}) d\psi_{1:50}$$

We do not make use of factorized joint distribution $q(z, \psi)$ to explore bound's behavior in high dimensions. We use the proposed bound from Theorem C.1 and compare it to SIVI (Yin and Zhou, 2018) on the task of upper-bounding the negative differential entropy $\mathbb{E}_{q(z)} \log q(z)$. For IWHVI we take $\tau(\psi \mid z)$ to be a Gamma distribution whose concentration and rate are generated by a neural

| Method | MNIST | OMNIGLOT |
|---|---|---|
| From (Mescheder et al., 2017) | | |
| AVB + AC | $-83.7 \pm 0.3$ | — |
| **IWHVI** | $-83.9 \pm 0.1$ | $-104.8 \pm 0.1$ |
| SIVI | $-84.4 \pm 0.1$ | $-105.7 \pm 0.1$ |
| HVM | $-84.9 \pm 0.1$ | $-105.8 \pm 0.1$ |
| VAE + RealNVP | $-84.8 \pm 0.1$ | $-106.0 \pm 0.1$ |
| VAE + IAF | $-84.9 \pm 0.1$ | $-107.0 \pm 0.1$ |
| VAE | $-85.0 \pm 0.1$ | $-106.6 \pm 0.1$ |

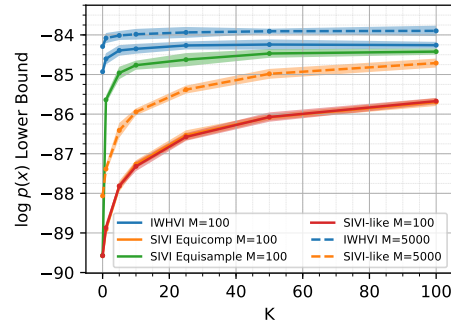

Figure 2: **Left**: Test log-likelihood on dynamically binarized MNIST and OMNIGLOT. **Right**: Comparison of multisample DIWHVI and SIVI-IW on a trained MNIST VAE from section 6.2 for $M = 100$ and 5000. Shaded area denotes $\pm 2$ std. interval, computed over 10 independent runs for each value of $K$.

network with three 500-dimensional hidden layers from $z$. We use the freedom to design architecture and initialize the network at prior. Namely, we also add a sigmoid "gate" output with large initial negative bias and use the gate to combine prior concentration and rate with those generated by the network. This way we are guaranteed to perform no worse than SIVI even at a randomly initialized $\tau$. Figure 1a shows the value of the bound for a different number of optimization steps over $\tau$ parameters, minimizing the bound. The whole process (including random initialization of neural networks) was repeated 50 times to compute empirical 90% confidence intervals. As results clearly indicate, the proposed bound can be made much tighter, more than halving the gap to the true negative entropy compared to SIVI and HVM.

## 6.2 Variational Autoencoder

We further test our method on the task of generative modeling, applying it to VAE (Kingma and Welling, 2013), which is a standard benchmark for inference methods [4]. Ideally, better inference should allow one to learn more expressive generative models. We report results on two datasets: MNIST (LeCun et al., 1998) and OMNIGLOT (Lake et al., 2015). For MNIST we follow the setup by Mescheder et al. (2017), and for OMNIGLOT we follow the standard setup (Burda et al., 2015). For experiment details see appendix G.

During training we used the proposed bound eq. (4) with standard prior $p(z) = \mathcal{N}(z \mid 0, 1)$ with increasing number $K$: we used $K = 0$ for the first 250 epochs, $K = 5$ for the next 250 epochs, and $K = 25$ for the next 500 epochs, and $K = 50$ from then on (90% of training). Such schedule is motivated by a natural observation (see the last paragraph of this section) that larger values of $K$ lead to more expressive variational models, yet large values of $K$ sometimes caused instabilities early in training due to an unlucky initialization. Regarding the number of samples $z$, we used $M = 1$ throughout training.

To estimate the log marginal likelihood for hierarchical models (IWHVI, SIVI, HVM) we use the DIWHVI lower bound (5) for $M = 5000$, $K = 100$ (for justification of DIWHVI as an evaluation metric see section 6.3). Results are shown in fig. 2. To evaluate the SIVI using the DIWHVI bound we fit $\tau$ to a trained model by making 7000 epochs on the trainset with $K = 50$, keeping parameters of $q_\phi(z, \psi \mid x)$ and $p_\theta(x, z)$ fixed. We observed improved performance compared to special cases of HVM and SIVI, and the method showed comparable results to the prior works.

For HVM on MNIST we observed its $\tau(\psi \mid z)$ essentially collapsed to $q(\psi)$, having expected KL divergence between the two extremely close to zero. This indicates the "posterior collapse" (Kim et al., 2018; Chen et al., 2016) problem where the inference network $q(z \mid \psi)$ choses to ignore the extra input $\psi$ and thus the whole model effectively degenerates to a vanilla VAE. At the same time IWHVI does not suffer from this problem due to non-zero $K$, achieving average $D_{KL}(\tau(\psi \mid z, x) \mid\mid q(\psi))$

of approximately 6.2 nats, see section 6.3. On OMNIGLOT the HVM did learn useful $\tau$ and achieved average $D_{KL}(\tau(\psi \mid z, x) \mid\mid q(\psi)) \approx 1.98$ nats, however the IWHVI did much better and achieved $\approx 9.97$ nats.

We also tried to learn hierarchical approximate posterior $q(z|x)$ using UIVI (Titsias and Ruiz, 2018). Unfortunately, the default parameters used in the original paper did not lead to a significant improvement over the standard VAE (-84.9 vs -85.0 on MNIST). We hypothesize this is due to HMC's poor mixing over different modes and high sensitivity to hyperparameters. This, combined with computational expensiveness of UIVI (our TensorFlow-based implementation was nearly 10 times slower than IWHVI, see discussion in section 5), prevented us from exploiting the hyperparameter space more exhaustively.

**Influence of** $K$: to investigate $K$'s influence on the training process, we trained VAEs on MNIST for 3000 epochs for different values of $K$ with SIVI, IWHVI and IWHVI-DReG, evaluated DIWHVI bound for $M = 1000, K = 100$ and Mutual Information (MI, see appendix F for details) between $z$ and $\psi$ under the joint $q_\phi(z, \psi|x)$. Results in fig. 1b clearly show that larger values of $K$ lead to better final models in terms of the log marginal likelihood, as well as approximate posteriors $q_\phi(z|x)$ that rely on the latent $\psi$ more heavily, as measured by the MI. Notably, the IWHVI achieves much higher values of the Mutual Information than the SIVI, and the improved gradient estimator IWHVI-DReG enables even better results due to better auxiliary distribution $\tau_\eta(\psi|z, x)$ (Tucker et al., 2019). These results empirically validate the claim that tighter bounds reduce the (auxiliary) variational bias.

## 6.3   DIWHVI as Evaluation Metric

One of the established approaches to evaluate the intractable log marginal likelihood in Latent Variable Models is to compute the multisample IWAE-bound with large $M$ since it is shown to converge to the log marginal likelihood as $M$ goes to infinity. Since both IWHVI and SIVI allow tightening the bound by taking more samples $z_m$, we compare methods along this direction.

Both DIWHVI and SIVI (being a special case of the former) can be shown to converge to log marginal likelihood as both $M$ and $K$ go to infinity, however, rates might differ. We empirically compare the two by evaluating an MNIST-trained IWHVAE model from section 6.2 for several different values $K$ and $M$. We use the proposed DIWHVI bound (5), and compare it with several SIVI modifications. We call *SIVI-like* the (5) with $\tau(\psi \mid z) = q(\psi)$, but without $\psi$ reuse, thus using $MK$ independent samples. *SIVI Equicomp* stands for sample reusing bound (3), which uses only $M + K$ samples, and uses same $\psi_{1:K}$ for every $z_m$. *SIVI Equisample* is a fair comparison in terms of the number of samples: we take $M(K + 1)$ samples of $\psi$, and reuse $MK$ of them for every $z_m$. This way we use the same number of samples $\psi$ as DIWHVI does, but perform $O(M^2 K)$ log-density evaluations to estimate $\log q(z \mid x)$, which is why we only examine the $M = 100$ case.

Results shown in section 6.2 indicate superior performance of the DIWHVI bound. Surprisingly SIVI-like and SIVI Equicomp estimates nearly coincide, with no significant difference in variance; thus we conclude sample reuse does not hurt SIVI. Still, there is a considerable gap to the IWHVI bound, which uses similar to SIVI-like amount of computing and samples. In a more fair comparison to the Equisample SIVI bound, the gap is significantly reduced, yet IWHVI is still a superior bound, especially in terms of computational efficiency, as there are no $O(M^2 K)$ operations.

Comparing IWHVI and SIVI-like for $M = 5000$ we see that the former converges after a few dozen samples, while SIVI is rapidly improving, yet lagging almost 1 nat behind for 100 samples, and even 0.5 nats behind the HVM bound (IWHVI for $K = 0$). One explanation for the observed behaviour is large $\mathbb{E}_{q(z|x)} D_{KL}(q(\psi \mid x) \mid\mid q(\psi \mid x, z))$, which was estimated [5] (on a test set) to be at least $46.85$ nats, causing many samples from $q(\psi)$ to generate poor likelihood $q(z \mid \psi)$ for a given $z_m$ due to large difference with the true inverse model $q(\psi \mid x, z)$. This is consistent with motivation layed out in section 2.1: a better approximate inverse model $\tau$ leads to more efficient sample usage. At the same time $\mathbb{E}_{q(z|x)} D_{KL}(\tau(\psi \mid x, z) \mid\mid q(\psi \mid x, z))$ was estimated to be approximately 3.24 and $\mathbb{E}_{q(z|x)} D_{KL}(\tau(\psi \mid x, z) \mid\mid q(\psi \mid x)) \approx 6.25$, proving that one can indeed do much better by learning $\tau(\psi \mid x, z)$ instead of using the prior $q(\psi \mid x)$.

# 7 Conclusion

We presented a multisample variational upper bound on the log marginal density, which allowed us to give tight tractable lower bounds on the intractable ELBO in the case of hierarchical variational model $q_\phi(z \mid x)$. We experimentally validated the bound and showed it alleviates (auxiliary) variational bias to a further extent than prior works do (which we showed to be a special cases of the proposed bound in appendix A), allowing for more expressive approximate posteriors, which does translate into a better inference. We then combined our bound with multisample IWAE bound, which led to a tighter lower bound of the log marginal likelihood. We therefore believe the proposed variational inference method will be useful for many approximate inference problems, and the multisample variational upper bound on log marginal density is a useful theoretical tool, allowing, for example, to give an upper bound on KL-divergence (appendix B) or to give sandwich bounds on the Mutual Information (appendix F).

## Acknowledgements

Authors would like to thank Aibek Alanov, Dmitry Molchanov, Oleg Ivanov and Anonymous Reviewer 3 for valuable discussions and feedback. Results on multisample extensions, shown in Section 4 have been obtained by Dmitry Vetrov and are supported by the Russian Science Foundation grant no.~17-71-20072.

## Footnotes

*Samsung-HSE Laboratory, National Research University Higher School of Economics

[2] One could also include all $\psi_{1:M,0}$ into the set of reused samples $\psi$, expanding its size to $M + K$.

[3]This choice makes the bound $\mathcal{U}_K$ equal to the log marginal density.

[4] Code is available at https://github.com/artsobolev/IWHVI

[5]Difference between $K$-sample IWAE and ELBO gives a lower bound on $D_{KL}(\tau(\psi \mid z) \mid\mid q(\psi \mid z))$, we used $K = 5000$.

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
