[Supplementary Material]

## A  Special Cases

In the appendix we will be considering the IWHVI bound it its full generality by assuming a hierarchical prior $p(z) = \int p(z, \zeta)d\zeta$:

$$\log p(x) \geq \mathop{\mathbb{E}}_{q(z|x)} \log \frac{p(x,z)}{q(z \mid x)} \geq \mathop{\mathbb{E}}_{q(z,\psi_0|x)} \mathop{\mathbb{E}}_{\tau(\psi_{1:K}|z,x)} \mathop{\mathbb{E}}_{\rho(\zeta_{1:L}|z)} \log \frac{p(x \mid z)\frac{1}{L}\sum_{k=1}^{L}\frac{p(z,\zeta_l)}{\rho(\zeta_l|z)}}{\frac{1}{K+1}\sum_{k=0}^{K}\frac{q(z,\psi_k|x)}{\tau(\psi_k|z,x)}} \quad (6)$$

This bound is obtained by a simple application of the IWAE bound to $\log p(z)$ term. Many previously known methods can be seen as special cases of this version of the bound. In particular:

- For an arbitrary $K, L, \tau(\psi \mid z, x) = q(\psi \mid x)$ (the hyperprior on $\psi$ under $q$) and $\rho(\zeta \mid z) = p(\zeta)$ we recover the DSIVI bound (Molchanov et al., 2018)

$$\log p(x) \geq \mathop{\mathbb{E}}_{q(z,\psi_0|x)} \mathop{\mathbb{E}}_{q(\psi_{1:K}|x)} \mathop{\mathbb{E}}_{p(\zeta_{1:K})} \log \frac{p(x \mid z)\frac{1}{L}\sum_{l=1}^{L}p(z \mid \zeta_l)}{\frac{1}{K+1}\sum_{k=0}^{K}q(z \mid \psi_k, x)}$$

- For an arbitrary $K$, $\tau(\psi \mid z, x) = q(\psi \mid x)$ and an explicit prior $p(z)$ (equivalently, $\rho(\zeta \mid z) = p(\zeta \mid z)$) we recover the SIVI bound (Yin and Zhou, 2018)

$$\log p(x) \geq \mathop{\mathbb{E}}_{q(z,\psi_0|x)} \mathop{\mathbb{E}}_{q(\psi_{1:K}|x)} \log \frac{p(x,z)}{\frac{1}{K+1}\sum_{k=0}^{K}q(z|\psi_k, x)}$$

- For $K = 0$, arbitrary $\tau(\psi \mid z, x)$ and explicit prior $p(z)$ (equivalently, $\rho(\zeta \mid z) = p(\zeta \mid z)$) we recover the HVM bound (Ranganath et al., 2016), also known as auxiliary variables bound (Agakov and Barber, 2004; Salimans et al., 2015; Maaløe et al., 2016)

$$\log p(x) \geq \mathop{\mathbb{E}}_{q(z,\psi_0|x)} \log \frac{p(x,z)}{\frac{q(z,\psi_0|x)}{\tau(\psi_0|z,x)}}$$

- For $K = 0$, $\tau(\psi \mid z, x) = p(\psi \mid z, x)$ and structurally similar to $q(z \mid x)$ prior $p(z) = \int p(z, \psi)d\psi$ we recover the joint bound (Louizos et al., 2017)

$$\log p(x) \geq \mathop{\mathbb{E}}_{q(z,\psi_0|x)} \log \frac{p(x \mid z)p(z,\psi_0)}{q(z,\psi_0 \mid x)}$$

- For an arbitrary $K$, factorized inference and prior models $q(z, \psi \mid x) = q(z \mid x)q(\psi \mid x)$, $p(z, \zeta) = p(z)p(\zeta)$, optimal $\tau(\psi \mid z, x) = q(\psi \mid x)$ and $\rho(\zeta \mid z) = p(\zeta)$ we recover the standard ELBO

$$\log p(x) \geq \mathop{\mathbb{E}}_{q(z|x)} \log \frac{p(x,z)}{q(z \mid x)}$$

  So even if there's no hierarchical structure, the bound still works.

## B  Sandwich Bounds on KL divergence between hierarchical models

Plain application of the bounds $\mathcal{U}_K$ and $\mathcal{L}_L$ to hierarchical $q(z)$ and $p(z)$ gives us the following upper bound on the KL divergence:

$$D_{KL}(q(z) \mid\mid p(z)) \leq \mathop{\mathbb{E}}_{q(z,\psi_0)} \mathop{\mathbb{E}}_{\tau(\psi_{1:K}|z)} \mathop{\mathbb{E}}_{\rho(\zeta_{1:L}|z)} \left[ \log \frac{\frac{1}{K+1}\sum_{k=0}^{K}\frac{q(z,\psi_k)}{\tau(\psi_k|z)}}{\frac{1}{L}\sum_{l=1}^{L}\frac{p(z,\zeta_l)}{\rho(\zeta_l|z)}} \right] \quad (7)$$

Similarly to (7) we can give a lower bound on KL divergence:

$$D_{KL}(q(z) \mid\mid p(z)) \geq \mathop{\mathbb{E}}_{q(z)} \mathop{\mathbb{E}}_{\tau(\psi_{1:K}|z)} \mathop{\mathbb{E}}_{p(\zeta_0|z)} \mathop{\mathbb{E}}_{\rho(\zeta_{1:L}|z)} \left[ \log \frac{\frac{1}{K}\sum_{k=1}^{K}\frac{q(z,\psi_k)}{\tau(\psi_k|z)}}{\frac{1}{L+1}\sum_{l=0}^{L}\frac{p(z,\zeta_l)}{\rho(\zeta_l|z)}} \right] \quad (8)$$

Unfortunately, this lower bound requires sampling from the true inverse $p(\zeta|z)$ and does not allow the same trick as (7), leaving us with expensive posterior sampling techniques. However, unlike the

previously suggested Fenchel conjugate based lower bound (Nowozin et al., 2016; Molchanov et al., 2018), the bound eq. (8) not only uses samples from $z$, but also makes use of its underlying density and can be made increasingly tighter by increasing $K$ and $L$.

Analyzing the gaps between the log marginal density $\log q_\phi(z \mid x)$ and the upper bound $\mathcal{U}_K$ (Theorem C.1) and the lower bound $\mathcal{L}_K$ (Domke and Sheldon, 2018), we see that by maximizing $\mathcal{L}_K$ or minimizing $\mathcal{U}_K$ we optimize different objectives w.r.t. $\tau$ (see Lemma C.2 for the definition of $\omega$):

$$\tau^* = \arg\min_{\tau \in \mathcal{F}} \mathcal{U}_K = \arg\min_{\tau \in \mathcal{F}} D_{KL}(q(\psi_0 \mid z)\tau(\psi_{1:K} \mid z) \parallel \omega_\tau(\psi_{0:K} \mid z))$$

$$\tau_-^* = \arg\max_{\tau \in \mathcal{F}} \mathcal{L}_K = \arg\min_{\tau \in \mathcal{F}} D_{KL}(\omega_\tau(\psi_{0:K} \mid z) \parallel q(\psi_0 \mid z)\tau(\psi_{1:K} \mid z))$$

In case $K = 0$ we have $\omega_\tau(\psi_{0:K} \mid x, z) = \tau(\psi_0 \mid x, z)$, and the gaps become simple forward and reverse KL divergences between $\tau(\psi \mid x, z)$ and the true inverse model $q(\psi \mid z, x)$. Thus unless $\tau$ (or $\rho$) is able to represent the true inverse model exactly, one should avoid using the same distribution in both log marginal density bounds and KL bounds (7) and (8).

# C  Proofs

**Theorem C.1** (Log marginal density upper bound). *For any $q(z, \psi)$, $K \in \mathbb{N}_0$ and $\tau(\psi \mid z)$ such that for any $\psi$ s.t. $\tau(\psi \mid z) = 0$ we have $q(\psi, z) = 0$, consider*

$$\mathcal{U}_K = \mathbb{E}_{q(\psi_0|z)} \mathbb{E}_{\tau(\psi_{1:K}|z)} \log\left(\frac{1}{K+1} \sum_{k=0}^{K} \frac{q(z, \psi_k)}{\tau(\psi_k \mid z)}\right)$$

*where we write $\tau(\psi_{1:K} \mid z) = \prod_{k=1}^{K} \tau(\psi_k \mid z)$ for brevity. Then the following holds:*

1. *$\mathcal{U}_K \geq \log q(z)$*

2. *$\mathcal{U}_K \geq \mathcal{U}_{K+1}$*

3. *$\lim_{K \to \infty} \mathcal{U}_K = \log q(z)$.*

*Proof.*    1. First, we note that an analogous proof of Molchanov et al. (2018) for Semi-Implicit VI can not be applied in our case as the argument of the $\log$ is no longer a valid mixture density in the case of arbitrary density $\tau$.

   Consider a gap between the proposed bound at the log marginal density:

$$\text{Gap} = \mathbb{E}_{q(\psi_0|z)} \mathbb{E}_{\tau(\psi_{1:K}|z)} \log\left(\frac{1}{K+1} \sum_{k=0}^{K} \frac{q(z, \psi_k)}{\tau(\psi_k \mid z)}\right) - \log q(z)$$

$$= \mathbb{E}_{q(\psi_0|z)} \mathbb{E}_{\tau(\psi_{1:K}|z)} \log\left(\frac{1}{K+1} \sum_{k=0}^{K} \frac{q(\psi_k \mid z)}{\tau(\psi_k \mid z)}\right)$$

$$= \mathbb{E}_{q(\psi_0|z)} \mathbb{E}_{\tau(\psi_{1:K}|z)} \log\left(\frac{q(\psi_0 \mid z)\tau(\psi_{1:K} \mid z)}{\omega_{q,\tau}(\psi_{0:K} \mid z)}\right)$$

$$= D_{KL}(q(\psi_0 \mid z)\tau(\psi_{1:K} \mid z) \parallel \omega_{q,\tau}(\psi_{0:K} \mid z)) \geq 0$$

   Where the last line holds due to $\omega_q$ being a normalized density function (see Lemma C.2):

$$\omega_{q,\tau}(\psi_{0:K} \mid z) = \frac{q(\psi_0 \mid z)\tau(\psi_{1:K} \mid z)}{\frac{1}{K+1} \sum_{k=0}^{K} \frac{q(\psi_k|z)}{\tau(\psi_k|z)}}$$

2. Now we will prove the second claim.

$$\mathcal{U}_K - \mathcal{U}_{K+1} = \underset{q(\psi_0|z)}{\mathbb{E}} \underset{\tau(\psi_{1:K+1}|z)}{\mathbb{E}} \log \frac{\frac{1}{K+1} \sum_{k=0}^{K} \frac{q(z,\psi_k)}{\tau(\psi_k|z)}}{\frac{1}{K+2} \sum_{k=0}^{K+1} \frac{q(z,\psi_k)}{\tau(\psi_k|z)}}$$

$$= \underset{q(\psi_0|z)}{\mathbb{E}} \underset{\tau(\psi_{1:K+1}|z)}{\mathbb{E}} \log \frac{q(\psi_0 \mid z)\tau(\psi_{1:K+1} \mid z)}{\nu_{q,\tau}(\psi_{0:K+1} \mid z)}$$

$$= D_{KL}\left(q(\psi_0 \mid z)\tau(\psi_{1:K+1} \mid z) \| \nu_{q,\tau}(\psi_{0:K+1} \mid z)\right) \geq 0$$

Where we used the fact that $\nu_{q,\tau}(\psi_{0:K+1} \mid z)$ is normalized density due to Lemma C.3

$$\nu_{q,\tau}(\psi_{0:K+1} \mid z) = \omega_{q,\tau}(\psi_{0:K} \mid z)\tau(\psi_{K+1} \mid z)\frac{1}{K+2} \sum_{k=0}^{K+1} \frac{q(\psi_k \mid z)}{\tau(\psi_k \mid z)}$$

3. For the last claim we follow (Burda et al., 2015). Consider

$$M_K = \frac{1}{K+1} \sum_{k=0}^{K} \frac{q(z,\psi_k)}{\tau(\psi_k \mid z)} = \overbrace{\frac{1}{K+1} \frac{q(z,\psi_0)}{\tau(\psi_0 \mid z)}}^{A_K} + \overbrace{\frac{K}{K+1}}^{B_K} \overbrace{\frac{1}{K} \sum_{k=1}^{K} \frac{q(z,\psi_k)}{\tau(\psi_k \mid z)}}^{X_K}$$

Due to Law of Large Numbers we have

$$A_K \xrightarrow[K\to\infty]{a.s.} 0, \qquad X_K \xrightarrow[K\to\infty]{a.s.} \underset{\tau(\psi|z)}{\mathbb{E}} \frac{q(z,\psi)}{\tau(\psi \mid z)} = q(z), \qquad B_K \xrightarrow[K\to\infty]{a.s.} 1$$

Thus

$$M_K \xrightarrow[K\to\infty]{a.s.} q(z), \qquad \mathcal{U}_K = \underset{\tau(\psi_{0:K}|z)}{\mathbb{E}} \log M_K \xrightarrow[K\to\infty]{} \log q(z)$$

$\square$

**Lemma C.2** ($\omega_{q,\tau}$ distribution, following Domke and Sheldon (2018)). *Given z, consider a following generative process:*

- *Sample $K+1$ i.i.d. samples from $\hat{\psi}_k \sim \tau(\psi \mid z)$*

- *For each sample compute its weight $w_k = \frac{q(\hat{\psi}_k, z)}{\tau(\hat{\psi}_k|z)}$*

- *Sample $h \sim Cat\left(\frac{w_0}{\sum_{k=0}^{K} w_k}, \dots, \frac{w_K}{\sum_{k=0}^{K} w_k}\right)$*

- *Put h-th sample first, and then the rest: $\psi_0 = \hat{\psi}_h$, $\psi_{1:K} = \hat{\psi}_{\setminus h}$*

*Then the marginal density of $\psi_{0:K}$*

$$\omega_{q,\tau}(\psi_{0:K} \mid z) = \frac{q(\psi_0 \mid z)\tau(\psi_{1:K} \mid z)}{\frac{1}{K+1} \sum_{k=0}^{K} \frac{q(\psi_k|z)}{\tau(\psi_k|z)}}$$

*Proof.* The joint density for the generative process described above is

$$\omega_{q,\tau}(\hat{\psi}_{0:K}, h, \psi_{0:K} \mid z) = \tau(\hat{\psi}_{0:K} \mid z)\frac{w_h}{\sum_{k=0}^{K} w_k}\delta(\psi_0 - \hat{\psi}_h)\delta(\psi_{1:K} - \hat{\psi}_{\setminus h})$$

One can see that this is indeed a normalized density

$$\int \sum_{h=0}^{K} \left(\int \omega_\tau(\hat{\psi}_{0:K}, h, \psi_{0:K} \mid z)d\psi_{0:K}\right) d\hat{\psi}_{0:K} = \int \sum_{h=0}^{K} \tau(\hat{\psi}_{0:K} \mid z)\frac{w_h}{\sum_{k=0}^{K} w_k}d\hat{\psi}_{0:K}$$

$$= \int \tau(\hat{\psi}_{0:K} \mid z) \sum_{h=0}^{K} \frac{w_h}{\sum_{k=0}^{K} w_k}d\hat{\psi}_{0:K} = \int \tau(\hat{\psi}_{0:K} \mid z)d\hat{\psi}_{0:K} = 1$$

The marginal density $\omega_{q,\tau}(\psi_{0:K} \mid z)$ then is

$$\omega_{q,\tau}(\psi_{0:K} \mid z) = \int \sum_{h=0}^{K} \tau(\hat{\psi}_{0:K} \mid z)\frac{w_h}{\sum_{k=0}^{K} w_k}\delta(\psi_0 - \hat{\psi}_h)\delta(\psi_{1:K} - \hat{\psi}_{\backslash h})d\hat{\psi}_{0:K}$$

$$= (K+1)\int \tau(\hat{\psi}_{0:K} \mid z)\frac{w_0}{\sum_{k=0}^{K} w_k}\delta(\psi_0 - \hat{\psi}_0)\delta(\psi_{1:K} - \hat{\psi}_{1:K})d\hat{\psi}_{0:K}$$

$$= \int \tau(\hat{\psi}_{1:K} \mid z)\frac{q(z,\hat{\psi}_0)}{\frac{1}{K+1}\sum_{k=0}^{K} w_k}\delta(\psi_0 - \hat{\psi}_0)\delta(\psi_{1:K} - \hat{\psi}_{1:K})d\hat{\psi}_{0:K}$$

$$= \tau(\psi_{1:K} \mid z)\frac{q(z,\psi_0)}{\frac{1}{K+1}\sum_{k=0}^{K} \frac{q(\psi_k,z)}{\tau(\psi_k \mid z)}} = \frac{q(\psi_0 \mid z)\tau(\psi_{1:K} \mid z)}{\frac{1}{K+1}\sum_{k=0}^{K} \frac{q(\psi_k \mid z)}{\tau(\psi_k \mid z)}}$$

Where on the second line we used the fact that integrand is symmetric under the choice of $h$.

While the derivations above show a generative process underlying the distribution, one could also show directly that $\omega_{q,\tau}(\psi_{0:K} \mid z)$ integrates to 1:

$$\int \omega_{q,\tau}(\psi_{0:K} \mid z)d\psi_{0:K} = \int \frac{q(\psi_0 \mid z)\tau(\psi_{1:K} \mid z)}{\frac{1}{K+1}\sum_{k=0}^{K} \frac{q(\psi_k \mid z)}{\tau(\psi_k \mid z)}}d\psi_{0:K} = \int \frac{\frac{q(\psi_0 \mid z)}{\tau(\psi_0 \mid z)}\tau(\psi_{0:K} \mid z)}{\frac{1}{K+1}\sum_{k=0}^{K} \frac{q(\psi_k \mid z)}{\tau(\psi_k \mid z)}}d\psi_{0:K}$$

$$= \int \frac{\frac{q(\psi_j \mid z)}{\tau(\psi_j \mid z)}\tau(\psi_{0:K} \mid z)}{\frac{1}{K+1}\sum_{k=0}^{K} \frac{q(\psi_k \mid z)}{\tau(\psi_k \mid z)}}d\psi_{0:K} = \frac{1}{K+1}\sum_{j=0}^{K}\int \frac{\frac{q(\psi_j \mid z)}{\tau(\psi_j \mid z)}\tau(\psi_{0:K} \mid z)}{\frac{1}{K+1}\sum_{k=0}^{K} \frac{q(\psi_k \mid z)}{\tau(\psi_k \mid z)}}d\psi_{0:K}$$

$$= \int \frac{\frac{1}{K+1}\sum_{j=0}^{K}\frac{q(\psi_j \mid z)}{\tau(\psi_j \mid z)}\tau(\psi_{0:K} \mid z)}{\frac{1}{K+1}\sum_{k=0}^{K} \frac{q(\psi_k \mid z)}{\tau(\psi_k \mid z)}}d\psi_{0:K} = \int \tau(\psi_{0:K} \mid z)d\psi_{0:K} = 1$$

Where in the 3rd inequality we've exchanged $\psi_0$ and $\psi_j$, and in the 4th equality we've used the fact that different choices of $j$ lead to the same integral.

$\square$

**Lemma C.3.** *Let*

$$\nu_{q,\tau}(\psi_{0:K+1} \mid z) = \omega_{q,\tau}(\psi_{0:K} \mid z)\tau(\psi_{K+1} \mid z)\frac{1}{K+2}\sum_{k=0}^{K+1}\frac{q(\psi_k \mid z)}{\tau(\psi_k \mid z)}$$

*Then $\nu_{q,\tau}(\psi_{0:K+1} \mid z)$ is a normalized density.*

*Proof.* $\nu_{q,\tau}(\psi_{0:K+1} \mid z)$ is non-negative due to all the terms being non-negative. Now we'll show it integrates to 1 (colors denote corresponding terms):

$$\int \omega_{q,\tau}(\psi_{0:K} \mid z)\tau(\psi_{K+1} \mid z)\frac{1}{K+2}\sum_{k=0}^{K+1}\frac{q(\psi_k \mid z)}{\tau(\psi_k \mid z)}d\psi_{0:K+1}$$

$$= \frac{1}{K+2}\int \omega_{q,\tau}(\psi_{0:K} \mid z)\left[\sum_{k=0}^{K}\frac{q(\psi_k \mid z)}{\tau(\psi_k \mid z)} + \int \tau(\psi_{K+1} \mid z)\frac{q(\psi_{K+1} \mid z)}{\tau(\psi_{K+1} \mid z)}d\psi_{K+1}\right]d\psi_{0:K}$$

$$= \frac{1}{K+2}\left[\int \frac{q(\psi_0 \mid z)\tau(\psi_{1:K} \mid z)}{\frac{1}{K+1}\sum_{k=0}^{K} \frac{q(\psi_k \mid z)}{\tau(\psi_k \mid z)}}\sum_{k=0}^{K}\frac{q(\psi_k \mid z)}{\tau(\psi_k \mid z)}d\psi_{0:K} + 1\right] = \frac{K+1+1}{K+2} = 1$$

$\square$

**Theorem C.4** (DIWHVI Evidence Lower Bound).

$$\log p(x) \geq \mathbb{E} \log \left[ \frac{1}{M} \sum_{m=1}^{M} \frac{p(x \mid z_m) \frac{1}{K} \sum_{k=1}^{K} \frac{p(z_m, \zeta_{m,k})}{\rho(\zeta_{m,k} \mid z_m)}}{\frac{1}{K+1} \sum_{k=0}^{K} \frac{q(z_m, \psi_{m,k} \mid x)}{\tau(\psi_{m,k} \mid z_m, x)}} \right] \tag{9}$$

*Where the expectation is taken over the following generative process:*

1. *Sample* $\psi_{m,0} \sim q(\psi \mid x)$ *for* $1 \leq m \leq M$

2. *Sample* $z_m \sim q(z \mid x_n, \psi_{m,0})$ *for* $1 \leq m \leq M$

3. *Sample* $\psi_{m,k} \sim \tau(\psi \mid z_m, x)$ *for* $1 \leq m \leq M$ *and* $1 \leq k \leq K$

4. *Sample* $\zeta_{m,k} \sim \rho(\zeta \mid z_m)$ *for* $1 \leq m \leq M$ *and* $1 \leq k \leq K$

*Proof.* Consider a random variable

$$X_M = \frac{1}{M} \sum_{m=1}^{M} \frac{p(x \mid z_m) \frac{1}{K} \sum_{k=1}^{K} \frac{p(z_m, \zeta_{m,k})}{\rho(\zeta_{m,k} \mid z_m)}}{\frac{1}{K+1} \sum_{k=0}^{K} \frac{q(z_m, \psi_{m,k} \mid x)}{\tau(\psi_{m,k} \mid z_m, x)}}$$

We'll show it's an unbiased estimate of $p(x)$ (colors denote corresponding terms) and then just invoke Jensen's inequality:

$$\mathbb{E} X_M = \int \left[ \left( \prod_{m=1}^{M} q(\psi_{m,0} \mid x) q(z_m \mid \psi_{m,0}, x) \tau(\psi_{m,1:K} \mid z_m, x) \rho(\zeta_{m,1:K} \mid z_m) \right) \right.$$
$$\left. \frac{1}{M} \sum_{m=1}^{M} \frac{p(x \mid z_m) \frac{1}{K} \sum_{k=1}^{K} \frac{p(z_m, \zeta_{m,k})}{\rho(\zeta_{m,k} \mid z_m)}}{\frac{1}{K+1} \sum_{k=0}^{K} \frac{q(z_m, \psi_{m,k} \mid x)}{\tau(\psi_{m,k} \mid z_m, x)}} \right] d\psi_{1:M,0:K} d\zeta_{1:M,1:K} dz_{1:M}$$

First, we move in the integral w.r.t. $\zeta$ into the numerator:

$$= \int \left( \frac{1}{M} \sum_{m=1}^{M} \frac{p(x \mid z_m) \underset{\rho(\zeta_{m,1:K} \mid z_m)}{\mathbb{E}} \frac{1}{K} \sum_{k=1}^{K} \frac{p(z_m, \zeta_{m,k})}{\rho(\zeta_{m,k} \mid z_m)}}{\frac{1}{K+1} \sum_{k=0}^{K} \frac{q(z_m, \psi_{m,k} \mid x)}{\tau(\psi_{m,k} \mid z_m, x)}} \right.$$
$$\left. \prod_{m=1}^{M} q(\psi_{m,0} \mid x) q(z_m \mid \psi_{m,0}, x) \tau(\psi_{m,1:K} \mid z_m, x) d\psi \right) dz$$

Next, we leverage $z_m$'s independence:

$$= \int \left[ \left( \prod_{m=1}^{M} q(\psi_{m,0} \mid x) q(z_m \mid \psi_{m,0}, x) \tau(\psi_{m,1:K} \mid z_m, x) \right) \right.$$
$$\left. \frac{1}{M} \sum_{m=1}^{M} \frac{p(x \mid z_m) p(z_m)}{\frac{1}{K+1} \sum_{k=0}^{K} \frac{q(z_m, \psi_{m,k} \mid x)}{\tau(\psi_{m,k} \mid z_m, x)}} \right] d\psi_{1:M,0:K} dz_{1:M}$$
$$= \frac{1}{M} \sum_{m=1}^{M} \int p(x, z_m) \frac{q(z_m, \psi_{m,0} \mid x) \tau(\psi_{m,1:K} \mid z_m)}{\frac{1}{K+1} \sum_{k=0}^{K} \frac{q(z_m, \psi_{m,k} \mid x)}{\tau(\psi_{m,k} \mid z_m, x)}} d\psi_{m,0:K} dz_m$$
$$= \frac{1}{M} \sum_{m=1}^{M} \int p(x, z_m) \omega_{q,\tau}(\psi_{m,0:K} \mid z_m, x) d\psi_{m,0:K} dz_m$$
$$= \frac{1}{M} \sum_{m=1}^{M} \int p(x, z_m) dz_m = p(x)$$

Where $\omega_{q,\tau}(\psi_{m,0:K} \mid z_m, x)$ is a density from the Lemma C.2. Now the rest follows from the Jensen's inequality due to logarithm's concavity:

$$\log p(x) = \log \mathbb{E} \, X_M \geq \mathbb{E} \log X_M$$

$\square$

**Corollary C.4.1.** *All statements of Theorem 1 of (Burda et al., 2015) apply to this bounds as well.*

## D   Signal-to-Noise Ratio Study

In this section we provide additional details to section 4.2.

### D.1   Doubly Reparametrized Gradient Derivation

Consider a VAE setup with multisample bound (5). Learning in such a model is equivalent to maximizing the following objective w.r.t. generative model's parameters $\theta$, inference network's parameters $\phi$, and auxiliary inference network $\tau$'s parameters $\eta$:

$$\mathcal{L}(\theta, \phi, \eta) = \underset{\substack{q_\phi(\psi_{1:M,0}, z_{1:M}|x) \\ \tau_\eta(\psi_{1:M,1:K}|x,z_{1:M})}}{\mathbb{E}} \log \frac{1}{M} \sum_{m=1}^{M} \frac{p_\theta(x, z_m)}{\frac{1}{K+1} \sum_{k=0}^{K} \frac{q_\phi(z_m, \psi_{m,k}|x)}{\tau_\eta(\psi_{m,k}|x,z_m)}}$$

$$= \underset{\substack{q_\phi(\psi_{1:M,0}, z_{1:M}|x) \\ \tau_\eta(\psi_{1:M,1:K}|x,z_{1:M})}}{\mathbb{E}} \underset{m=1}{\overset{M}{\mathrm{L\Sigma E}}} \left[ \overbrace{\log p_\theta(x \mid z_m) - \underset{k=0}{\overset{K}{\mathrm{L\Sigma E}}} \left( \underbrace{\log \frac{q_\phi(z_m, \psi_{m,k}|x)}{p_\theta(z_m)\tau_\eta(\psi_{m,k}|x,z_m)}}_{\beta_{mk}} \right)}^{\alpha_m} \right] + \mathrm{const}$$

Where $\mathrm{L\Sigma E}$ is a shorthand for the log-sum-exp operator, and the omitted constant is $\log \frac{K+1}{M}$. We will now consider a *reparametrized* gradient w.r.t. $\tau$'s parameters $\nabla_\eta \mathcal{L}$, notice the REINFORCE-like term still sitting inside, and carve it out by applying the reparametrization the second time in the same way as in (Tucker et al., 2019). In the derivation we'll use $\boldsymbol{\sigma}$ notation to mean the softmax $\frac{1}{1+\exp(-x)}$ function, and $\nabla_\eta \psi$ is a shorthand for $(\nabla_\eta g(\varepsilon, \eta))|_{\varepsilon=g^{-1}(\psi,\eta)}$ where $g$ is $\tau_\eta(\psi \mid x, z)$'s reparametrization.

$$\nabla_\eta \mathcal{L}(\theta, \phi, \eta) = \underset{q_\phi(\psi_{1:M,0}, z_{1:M})}{\mathbb{E}} \nabla_\eta \underset{\tau_\eta(\psi_{1:M,1:K}|z_{1:M})}{\mathbb{E}} \underset{m=1}{\overset{M}{\mathrm{L\Sigma E}}}[\alpha_m]$$

$$= \underset{q_\phi(\psi_{1:M,0}, z_{1:M})}{\mathbb{E}} \left( \underset{\tau_\eta(\psi_{1:M,1:K}|z_{1:M})}{\mathbb{E}} \nabla_\eta \underset{m=1}{\overset{M}{\mathrm{L\Sigma E}}}[\alpha_m] \right.$$

$$\left. + \underset{\tau_\eta(\psi_{1:M,1:K}|z_{1:M})}{\mathbb{E}} \sum_{m=1}^{M} \sum_{k=1}^{K} \nabla_{\psi_{mk}} \left( \underset{m=1}{\overset{M}{\mathrm{L\Sigma E}}}[\alpha_m] \right) \nabla_\eta \psi_{mk} \right)$$

$$= \underset{q_\phi(\psi_{1:M,0}, z_{1:M})}{\mathbb{E}} \left( \underset{\tau_\eta(\psi_{1:M,1:K}|z_{1:M})}{\mathbb{E}} \sum_{m=1}^{M} \boldsymbol{\sigma}(\alpha)_m \nabla_\eta \alpha_m + A \right)$$

$$= \underset{q_\phi(\psi_{1:M,0}, z_{1:M})}{\mathbb{E}} \left( \underset{\tau_\eta(\psi_{1:M,1:K}|z_{1:M})}{\mathbb{E}} \sum_{m=1}^{M} \boldsymbol{\sigma}(\alpha)_m \left[ -\sum_{k=0}^{K} \boldsymbol{\sigma}(\beta_m)_k \nabla_\eta \beta_{mk} \right] + A \right)$$

$$= \underset{q_\phi(\psi_{1:M,0}, z_{1:M})}{\mathbb{E}} \left( \underset{\tau_\eta(\psi_{1:M,1:K}|z_{1:M})}{\mathbb{E}} \sum_{m=1}^{M} \boldsymbol{\sigma}(\alpha)_m \left[ \sum_{k=0}^{K} \boldsymbol{\sigma}(\beta_m)_k \nabla_\eta \log \tau_\eta(\psi_{mk}) \right] + A \right)$$

$$= \underset{q_\phi(\dots)}{\mathbb{E}} \left( \underset{\tau_\eta(\dots)}{\mathbb{E}} \sum_{m=1}^{M} \sum_{k=1}^{K} \overbrace{\textcolor{red}{\boldsymbol{\sigma}(\alpha)_m \, \boldsymbol{\sigma}(\beta_m)_k \nabla_\eta \log \tau_\eta(\psi_{mk})}}^{\text{REINFORCE-like term}} + B + A \right)$$

Where

$$A = - \underset{\tau_\eta(\psi_{1:M,1:K}|z_{1:M})}{\mathbb{E}} \sum_{m=1}^{M} \sum_{k=1}^{K} \boldsymbol{\sigma}(\alpha)_m \, \boldsymbol{\sigma}(\beta_m)_k \nabla_{\psi_{mk}} \beta_{mk} \nabla_\eta \psi_{mk}$$

$$B = \underset{\tau_\eta(\psi_{1:M,1:K}|z_{1:M})}{\mathbb{E}} \sum_{m=1}^{M} \boldsymbol{\sigma}(\alpha)_m \, \boldsymbol{\sigma}(\beta_m)_0 \nabla_\eta \log \tau_\eta(\psi_{m0})$$

$$q_\phi(\dots) = q_\phi(\psi_{1:M,0}, z_{1:M}), \qquad \tau_\eta(\dots) = \tau_\eta(\psi_{1:M,1:K} \mid z_{1:M})$$

One can see the red term is the REINFORCE derivative of the $\mathbb{E}_{\tau_\gamma(\psi_{m,k})} \boldsymbol{\sigma}(\alpha)_m \, \boldsymbol{\sigma}(\beta_m)_k$ w.r.t. $\gamma$ with all other $\psi$ being fixed. We then evaluate this REINFORCE gradient at $\gamma = \eta$, this substitution "trick" is needed to avoid differentiating $\alpha$ and $\beta$ w.r.t. $\eta$, only their gradients w.r.t. $\psi_{mk}$ matter. We would also like to notice that though $B$ also contains $\nabla_\eta \log \tau_\eta(\psi_{m0})$ term similar to REINFORCE, it's not a REINFORCE gradient, as $\psi_{m0}$ comes from a different distribution, and thus we can not apply reparametrization to it.

$$
\begin{aligned}
\nabla_\eta \mathcal{L}(\theta, \phi, \eta) &= \underset{q_\phi(\dots)}{\mathbb{E}} \left( \sum_{m=1}^{M} \sum_{k=1}^{K} \underset{\tau_\eta(\psi_{1:M,1:K\setminus mk}|z_{1:M})}{\mathbb{E}} \left( \nabla_\gamma \mathbb{E}_{\tau_\gamma(\psi_{mk}|z_m)} \boldsymbol{\sigma}(\alpha)_m \, \boldsymbol{\sigma}(\beta_m)_k \right) \Big|_{\gamma=\eta} + B + A \right) \\
&= \underset{q_\phi(\dots)}{\mathbb{E}} \left( \sum_{m=1}^{M} \sum_{k=1}^{K} \mathbb{E}_{\tau_\eta(\psi_{1:M,1:K}|z_{1:M})} \nabla_{\beta_{mk}} [\boldsymbol{\sigma}(\alpha)_m \, \boldsymbol{\sigma}(\beta_m)_k] \nabla_{\psi_{mk}} \beta_{mk} \nabla_\eta \psi_{mk} + B + A \right) \\
&= \underset{\substack{q_\phi(\dots) \\ \tau_\eta(\dots)}}{\mathbb{E}} \left( \sum_{m=1}^{M} \sum_{k=1}^{K} \boldsymbol{\sigma}(\alpha)_m \, \boldsymbol{\sigma}(\beta_m)_k [1 - \boldsymbol{\sigma}(\beta_m)_k (2 - \boldsymbol{\sigma}(\alpha)_m)] \nabla_{\psi_{mk}} \beta_{mk} \nabla_\eta \psi_{mk} + B + A \right) \\
&= \underset{\substack{q_\phi(\dots) \\ \tau_\eta(\dots)}}{\mathbb{E}} \left( \sum_{m=1}^{M} \sum_{k=1}^{K} \boldsymbol{\sigma}(\alpha)_m \, \boldsymbol{\sigma}(\beta_m)_k^2 (\boldsymbol{\sigma}(\alpha)_m - 2) \nabla_{\psi_{mk}} \beta_{mk} \nabla_\eta \psi_{mk} + B \right) \\
&= \underset{\substack{q_\phi(\dots) \\ \tau_\eta(\dots)}}{\mathbb{E}} \sum_{m=1}^{M} \boldsymbol{\sigma}(\alpha)_m \left( (\boldsymbol{\sigma}(\alpha)_m - 2) \sum_{k=1}^{K} \boldsymbol{\sigma}(\beta_m)_k^2 \nabla_{\psi_{mk}} \beta_{mk} \nabla_\eta \psi_{mk} + \boldsymbol{\sigma}(\beta_m)_0 \nabla_\eta \log \tau_\eta(\psi_{m0}) \right)
\end{aligned}
$$

$$(10)$$

We call this gradient estimator IWHVI-DReG. Similar derivations can be made w.r.t. $q$'s parameters $\phi$ to combat decreasing signal-to-noise ratios identified by Rainforth et al. (2018). We will not provide them here, as this is outside of the scope of the present work. It's also straightforward to derive a similar gradient estimator for $\rho$'s parameters in case of hierarchical prior $p(z)$, since this case essentially corresponds to nested IWAE.

### D.2   Experiments

To evaluate the (10) gradient estimate we take a slightly trained (for 50 epochs by the usual gradient) VAE and compare signal-to-noise ratios of different gradients while varying $K$ and $M$. For each minibatch we recompute the vanilla autodiff gradients and a doubly reparametrized gradient (10) 100 times to estimate signal-to-noise ratio for each weight. We then average SNRs over different minibatches of fixed size, and present mean and 90% confidence intervals over different choices of weight in fig. 3a.

We also compute SNR on a toy task from (Rainforth et al., 2018) for an upper bound (theorem C.1) on $\mathbb{E}_{q(x)} \log q(x)$ for $q(x, z) = \mathcal{N}(x \mid z, I)\mathcal{N}(z \mid \theta, I)$ and $\tau_\eta(z \mid x) = \mathcal{N}(z \mid Ax + b, 2/3)$ for $\theta = [1, \dots, 1], x, z \in \mathbb{R}^{10}$. We first train $\eta = \{A, b\}$ to optimality by making 1000 AMSGrad (Reddi et al., 2018) steps with learning rate $10^{-2}$ with batches of 100, then evaluate the gradients 1000 times with batches of size 100. We also include gradients of IWAE (coming from the task of lower bounding the $\mathbb{E}_{q(x)} \log q(x)$) to compare with. We compute SNR per parameter over these 1000 samples, and present results in fig. 3b.

Best seen in the toy task, SNR of autodiff gradients decreases as $K$ grows, much resembling the standard IWAE issue, outlined by (Rainforth et al., 2018). IWHVI-DReG solves the problem, but

(a) IWHVI-VAE for $M = 1$ and $M = 25$                             (b) Toy Task

Figure 3: Signal-to-Noise Ratio of gradients for different values of $K$. Solid lines denote SNR averaged over all model's parameters, and shaded area marks 90% confidence interval over all parameters.

only partially. While the SNR is no longer decreasing, it does not increase as in IWAE-DReG. While detailed study of this phenomena is outside the scope of this work, we show that it's the second term (denoted $B$) in IWHVI-DReG, coming from the $\psi_{m0}$-based term in the bound, that causes the trouble. Indeed, if we omit this term, the (now biased) estimate would have higher SNR, essentially approaching the standard IWAE's one. In practice, we found that the improved gradient estimate IWHVI-DReG significantly increased $D_{KL}(\tau(\psi \mid z) \mid\mid q(z))$, but this resulted in minor increases in the validation log-likelihood.

# E   Debiasing the bound

## E.1   Deriving the bound

Following (Nowozin, 2018) we argue (lemma E.1) that

$$\mathcal{U}_K = \mathbb{E}_{p(\psi_0|z)} \mathbb{E}_{\tau(\psi_{1:K}|z)} \log \left( \frac{1}{K+1} \sum_{k=0}^{K} \frac{p(z, \psi_k)}{\tau(\psi_k \mid z)} \right) = \log p(z) + \sum_{j=1}^{\infty} \frac{\gamma_j}{(K+1)^j}$$

So $\mathcal{U}_K$ can be seen as a biased estimator of log marginal density $\log p(z)$ with bias of order $O(1/K)$. We can reduce this bias further by making use of the following fact:

$$(K+1)\mathcal{U}_K - K\mathcal{U}_{K-1} = \log p(z) - \frac{\gamma_2}{K(K+1)} + O\left(\frac{1}{K^2}\right)$$

$$= \log p(z) - \sum_{j=0}^{\infty} \frac{\gamma_2}{(K+1)^{j+2}} + O\left(\frac{1}{K^2}\right)$$

$$= \log p(z) - \frac{\gamma_2}{(K+1)^2} + O\left(\frac{1}{K^2}\right)$$

Averaging this identity over all possible choices of a subset of samples from $\tau(\psi \mid z)$ of size $K-1$, we obtain Jackknife-corrected estimator with bias of order $O(1/K^2)$. One could then apply the same procedure again and again to obtain Generalized-order-$J$-Jackknife-corrected estimator with bias of order $O(1/K^{J+1})$, leading to a $J$-Jackknife upper log marginal density estimate:

$$\mathcal{J}_K^J = \sum_{j=0}^{J} c(K, J, j) \overline{\mathcal{U}}_{K-j} \tag{11}$$

Where $\overline{\mathcal{U}}_{K-j}$ is $(K-j)$-samples bound averaged over all possible choices of a subset of size $K-j$ from $\psi_{1:K}$, and $c(K, J, j)$ are Sharot coefficients (Sharot, 1976; Nowozin, 2018):

$$\overline{\mathcal{U}}_{K-j} = \frac{1}{\binom{K}{K-j}} \sum_{S \subseteq \{1,\ldots,K\}:|S|=K-j} \log \left( \frac{1}{K-j+1} \left[ \frac{p(z, \psi_0)}{\tau(\psi_0 \mid z)} + \sum_{k \in S} \frac{p(z, \psi_k)}{\tau(\psi_k \mid z)} \right] \right)$$

$$c(K, J, j) = (-1)^j \frac{(K-j)^J}{(J-j)!j!}$$

Despite not guaranteed to be an upper bound anymore, we still see that the estimate tends to overestimate the log marginal density in practice.

**Lemma E.1.** *For fixed $p(\psi_0 \mid z), \tau(\psi_{1:K} \mid z)$ and $z$ there exists a sequence $\{\gamma_k\}_{k=1}^{\infty}$ s.t.*

$$\mathcal{U}_K = \log p(z) + \sum_{j=1}^{\infty} \frac{\gamma_j}{(K+1)^j} \tag{12}$$

*Proof.* First, we note that $\mathcal{U}_K$ can be represented as log marginal density plus some non-negative (due to theorem C.1) bias, which we'll consider in greater detail.

$$\mathcal{U}_K = \underset{\substack{p(\psi_0|z) \\ \tau(\psi_{1:K}|z)}}{\mathbb{E}} \log\left(\frac{1}{K+1}\sum_{k=0}^{K}\frac{p(z,\psi_k)}{\tau(\psi_k \mid z)}\right) = \log p(z) + \overbrace{\underset{\substack{p(\psi_0|z) \\ \tau(\psi_{1:K}|z)}}{\mathbb{E}} \log\left(\frac{1}{K+1}\sum_{k=0}^{K}\frac{p(\psi_k \mid z)}{\tau(\psi_k \mid z)}\right)}^{\text{Bias}}$$

Denote $w_k = \frac{p(\psi_k|z)}{\tau(\psi_k|z)}$, $w_k' = w_k - 1$ and expand the Bias around 1:

$$
\begin{aligned}
\text{Bias} &= \sum_{n=1}^{\infty} \frac{(-1)^{n+1}}{n} \underset{p(\psi_0|z)}{\mathbb{E}} \underset{\tau(\psi_{1:K}|z)}{\mathbb{E}} \left(\frac{1}{K+1}\sum_{k=0}^{K} w_k'\right)^n \\
&= \sum_{n=1}^{\infty} \frac{(-1)^{n+1}}{n} \underset{p(\psi_0|z)}{\mathbb{E}} \underset{\tau(\psi_{1:K}|z)}{\mathbb{E}} \left(\frac{1}{K+1}w_0' + \frac{K}{K+1}\overline{w_{1:K}'}\right)^n \\
&= \sum_{n=1}^{\infty} \frac{(-1)^{n+1}}{n} \sum_{m=0}^{n} \binom{n}{m} \underset{p(\psi_0|z)}{\mathbb{E}} \left(\frac{w_0'}{K+1}\right)^m \underset{\tau(\psi_{1:K}|z)}{\mathbb{E}} \left(\frac{K}{K+1}\overline{w_{1:K}'}\right)^{n-m} \\
&= \sum_{n=1}^{\infty} \frac{(-1)^{n+1}}{n} \sum_{m=0}^{n} \binom{n}{m} \left(1 - \frac{1}{K+1}\right)^{n-m} \left(\frac{1}{K+1}\right)^n \underset{p(\psi_0|z)}{\mathbb{E}} (w_0')^m \underset{\tau(\psi_{1:K}|z)}{\mathbb{E}} \left(\overline{w_{1:K}'}\right)^{n-m}
\end{aligned}
$$

Where $\overline{w_{1:K}'} = \frac{1}{K}\sum_{k=1}^{K} w_k'$ – an empirical average of zero-mean random variables. A. Angelova (2012) has shown that for a fixed $s \in \mathcal{N}$ there exist $\gamma_1^{(s)}, \ldots, \gamma_T^{(s)}$ s.t.

$$\underset{\tau(\psi_{1:K}|z)}{\mathbb{E}} \left(\overline{w_{1:K}'}\right)^s = \sum_{t=1}^{T} \frac{\gamma_t^{(s)}}{K^t} = \sum_{t=1}^{T} \gamma_t^{(s)} \left[\sum_{n=1}^{\infty} \frac{1}{(K+1)^n}\right]^t$$

Therefore every addend in Bias depends on $K$ only through $1/(K+1)$, and thus decomposition (12) holds.

Note on convergence: while we have not shown the presented series to be convergent, in practice we only care about bias' asymptotic behaviour up to an order of $1/(K+1)^J$, for which the decomposition works as long as the corresponding moments in Bias exist and are finite.

$\square$

## E.2 Experiments

We only perform experimental validation of the Jackknife upper KL estimate (JHVI for short) in the same setting as in section 6.1. appendix E.2 shows improved performance of the Jackknife-corrected estimate.

Figure 4: Negative entropy bound for 50-dimensional Laplace distribution. Shaded area denotes 90% confidence interval.

## F  Mutual Information

To estimate the Mutual Information (MI) of $q_\phi(z, \psi|x)$ with high certainty we rely on the multisample variational bounds $\mathcal{U}_K$ and $\mathcal{L}_K$. Indeed, the MI can be represented as:

$$\text{MI}[q_\phi(z, \psi|x)] := \mathop{\mathbb{E}}_{q_\phi(z,\psi|x)} \log \frac{q_\phi(z, \psi|x)}{q_\phi(z|x)q(\psi|x)} = \mathop{\mathbb{E}}_{q_\phi(z,\psi|x)} \left[\log q_\phi(z|\psi, x) - \log q_\phi(z|x)\right]$$

By applying $\mathcal{U}_K$ and $\mathcal{L}_K$ to the $\log q_\phi(z|x)$ term we obtain lower and upper bounds, correspondingly:

$$\mathop{\mathbb{E}}_{\substack{q_\phi(z,\psi_0|x) \\ \tau(\psi_{1:K}|z,x)}} \log \frac{q_\phi(z|\psi_0, x)}{\frac{1}{K+1}\sum_{k=0}^{K} \frac{q_\phi(z,\psi_k|x)}{\tau(\psi_k|x,z)}} \leq \text{MI}[q_\phi(z, \psi|x)] \leq \mathop{\mathbb{E}}_{\substack{q_\phi(z,\psi_0|x) \\ \tau(\psi_{1:K}|z,x)}} \log \frac{q_\phi(z|\psi_0, x)}{\frac{1}{K}\sum_{k=1}^{K} \frac{q_\phi(z,\psi_k|x)}{\tau(\psi_k|x,z)}}$$

These bounds can be made much tighter than previously known ones (Poole et al., 2018) by training better variational approximations and using many samples $K$, and in fact generalize and bridge the InfoNCE and Barber-Agakov lower bounds. In our evaluation, we used $K = 1000$. For SIVI models we fixed parameters $\theta$ and $\phi$ and trained only $\tau$ for 50 epochs on the trainset. While this probably wasn't enough to get the network to converge, we have used enough samples $K$ that the difference between the upper and the lower bounds become smaller than 0.1. The MI bounds were then averaged over the entire test set. We estimated the standard deviation of both bounds to be smaller than 0.01 for almost all models. The average of two bounds was used as the MI estimate.

## G  Experiments Details

For MNIST we follow the setup by Mescheder et al. (2017): we use single 32-dimensional stochastic layer with $p(z) = \mathcal{N}(z \mid 0, I)$ prior, decoder $p_\theta(x \mid z) = \text{Bernoulli}(x \mid \pi_\theta(z))$ where $\pi_\theta$ is a neural network with two hidden 300-neurons layers and a softplus nonlinearity [6], and latent variable model encoder $q_\phi(z \mid x) = \int \mathcal{N}(z \mid \mu_\phi(x, \psi), \sigma_\phi^2(x, \psi))\mathcal{N}(\psi \mid 0, 1)d\psi$ where $\mu_\phi(x, \psi)$ and $\sigma_\phi^2(x, \psi)$ are outputs of a neural network with architecture similar to as that of the decoder, except each next layer (including the one that generates distribution's parameters) acts on previous layer's output concatenated with input $\psi$. We take $\tau_\vartheta(\psi \mid z, x) = \mathcal{N}(\psi \mid \nu_\vartheta(x, z), \varsigma_\vartheta^2(x, z))$ where mean and variance are outputs of another neural network with same network architecture as that of the decoder, except it takes concatenation of $x$ and $z$ as input.

For OMNIGLOT we used simiar architecture, but for 50-dimensional $z$ and $\psi$, and all hidden layers had 200 neurons.

For flow-based models we used their implementations from TensorFlow Probability (Dillon et al., 2017a). We had the encoder network output not only $\mu$ and $\sigma$, but also a context vector $h$ of same size as $z$ to be used to condition the flow transformations.

For IWHVI we used a grid search over 1) whether to use IWHVI-DReG (10), 2) whether to do inner KL (the $\log \frac{q(\psi)}{\tau(\psi|x,z)}$ terms) warm-up (linearly increase its weight from 0 to 1) for the first 300 epochs, 3) whether to do outer KL warmup (the $\log \frac{p(z)}{\frac{1}{K+1}\sum\cdots}$ term) for the first 300 epochs, 4) Either use learning rate $\eta_{\text{base}} = 10^{-3}$ with annealing $\eta = 0.95^{\text{epoch}/100}\eta_{\text{base}}$ or $\eta = 10^{-4}$ without annealing. We used the same grid for SIVI, except options (1) and (2) had no effect, since they only influenced $\tau$, and thus we did not perform search over them. For HVM only the option (1) had no effect, since HVM does not sample from $\tau$.

In all cases using IWHVI-DReG proved beneficial, the optimal learning rate turned out to be 0.001 and outer KL warmup was useful. The inner KL warmup didn't affect the results significantly.

In all experiments we did $10,000$ epochs with batches of size 256 and used Adam (Kingma and Ba, 2014) optimizer with $\beta_1 = 0.9, \beta_2 = 0.999$. We held out $5,000$ examples from trainsets to be used as validation. The data was dynamically binarized by sampling each pixel as Bernoulli random variable with the probability of success being equal to the pixel's intensity.

# H    Software acknowledgements

This work would be impossible without numerous free high-quality scientific software packages. We are particularly grateful for TensorFlow (Abadi et al., 2015) and TensorFlow Probability (Dillon et al., 2017b), Numpy (Oliphant, 06 ), Scipy (Virtanen et al., 2019), matplotlib (Hunter, 2007).

## Footnotes

[6]This is the nonlinearity used in Mescheder et al. (2017)'s code for fully-connected experiments.