[Reviews · NeurIPS 2019]

Reviewer 1



After feedback: I stand by my original review and think it's a worthy submisson. The authors propose a novel lower bound on the ELBO for variational approximations defined by auxiliary random variables. The quality and clarity of the paper is high and the method is interesting. I however have a few comments: - Lines 56-59, the discussion regarding maximizing the ELBO wrt both sets of parameters was a bit unclear. Maximizing wrt to the variational parameters does minimize the KL, but since the KL depends also on the model parameters it is not necessarily the case that you are maximizing the log-marginal likelihood. Perhaps this was what the authors meant? - Lines 62-65. The result on the connection between self-normalized importance sampling and the ELBO for a more flexible variational approximation was first proved in Naesseth et al. (2018): Naesseth et al., Variational Sequential Monte Carlo, AISTATS, 2018. - The legend in some of the plots is obscuring some information (Fig. 1) - Line 26, "on" -> "in"?

Reviewer 2



There are some minor issues that can be improved. (1) The key idea of this work is not so clear. If my understanding is correct, the authors propose to use an importance weighted auxiliary variational distribution (See eq 1 & 4). I ignore the data x for notation simplicity in the following expressions. In Yin & Zhou 2018, a forward variational distribution is used such as q(z|\psi_0) \prod_{i=0} q(\psi_i) (see eq 2). In this work, the authors would like to use a backward variational distribution such as q(z) \prod_{i=0} q(\psi_i|z) = q(z|\psi_0)q(\psi_0)\prod_{i=1}q(\psi_i|z) . However, computing q(\psi_i|z) is intractable, the authors propose to use \tau(\psi_i|z) to approximate q(\psi_i|z). (see eq 4) I think the backward approximation (in other words \tau(\psi_i|z) should depend on z) gives the improvement over SIVI . The key building block for using the importance weight is Thm 1 at Domke & Sheldon 2018, where Domke & Sheldon consider an auxiliary distribution for the model p while in this work the authors use a similar idea of Thm 1 in the hierarchical variational model q such as Yin & Zhou 2018, and Molchanov et al 2018. (2) I think the proof of the Thm (C.1 in the appendix) in Section 3 is inspired by Molchanov et al 2018. If so, the authors should cite Molchanov et al 2018. (3) The proof of Lemma C.2 in the appendix can be simplified. I think it is easy to directly show that \omega (\psi_0:K |z) is normalized. The proof sketch is given below. Let C= \sum_{k=0}^K q(\psi_k|z) / \tau(\psi_k|z). We have \int \omega(\psi_0:K|z) d\psi_0:K = (K+1) \int q(\psi_0|z)/\tau(\psi_0|z) \tau(\psi_0:K|z) / C d\psi_0:K = \sum_{k=0}^{K} \int q(\psi_k|z)/tau(\psi_k|z) \tau(\psi_0:K|z) / C d\psi_0:K (due to the symmetry of these K+1 intergrals) = \int \sum_{k=0}^{K} q(\psi_k|z)/tau(\psi_k|z) \tau(\psi_0:K|z) / C d\psi_0:K = \int C \tau(\psi_0:K|z) / C d\psi_0:K = \int \tau(\psi_0:K|z) d\psi_0:K = 1

Reviewer 3



I have read the author response and have decided to raise my original score to a 6. Please do correct the erroneous statements regarding upper bounding the log marginal likelihood of the data and also regarding claim that maximizing the lower bound maximizes the log marginal likelihood of the data. Originality: The proposed method builds marginally upon SIVI [1] Clarity: Clarity is lacking...the abstract for example is obscure...what is the "marginal log-density" here? I believe it is the logarithm of q_{\phi}(z | x). Please make this clear at least once because the term "marginal log-density" is used throughout the paper. Quality: Quality is also lacking. For example the related work section does not mention some related work (see [2] and [3] below). Furthermore, there is a very misleading statement on line 205 about the core contribution of the paper: "the core contribution of the paper is a novel upper bound on marginal log-likelihood". This is a false statement. The paper proposes an upper bound of the log density of the variational distribution which leads to a lower bound (NOT an upper bound) on the log marginal likelihood of the data. Finally, the experiments don't include comparison to UIVI [4] which also improves upon SIVI and there is no report of computational time which would put the test log-likelihood improvement into perspective. Significance: I am not convinced the proposed method is significant. Questions and Minor Comments: 1--line 38 and elsewhere: it is "log marginal likelihood" not "marginal log-likelihood" 2--lines 57-59 are not clear...what do you mean here? 3--what is the intuition behind increasing K during training? How was the schedule for K chosen? [1] Semi-Implicit Variational Inference. Yin and Zhou, 2018. [2] Nonparametric Variational Inference. Gershman et al., 2012. [3] Hierarchical Implicit Models and Likelihood-Free Variational Inference. Tran et al., 2017. [4] Unbiased Implicit Variational Inference. Ruiz and Titsias, 2019.

[Author Response · NeurIPS 2019]

First, we thank all reviewers for their time and valuable feedback. We reply to each reviewer individually and then comment on the significance of the work.

To **Reviewer 1**: Yes, you understood the paragraph on lines 56-59 correctly. Optimizing the lower bound does not imply maximization of the log marginal likelihood; thus we should seek to close this gap as much as possible by choosing a flexible approximate posterior $q(z|x)$. If $q(z|x)$ is limited in its expressivity, the true posterior would also have to be simplistic, since the KL divergence gap forces the two to be close. We will rewrite this paragraph to make it more clear for the camera-ready version (This paragraph is also relevant to **Reviewer 4**).

We will add the missing citation of Naesseth et al. (2018) and fix the legend and the typo.

To **Reviewer 3**: yes, the key idea is to learn the auxiliary variational distribution $\tau(\psi|z)$ so that the samples $\psi_{1:K}$ are coming from the high-probability region of the optimal distribution $q(\psi|z)$. In contrast, SIVI uses samples from $q(\psi)$, which are uninformed about the particular $z$ and thus much more samples are needed to achieve the same quality, as we have shown empirically (please see the paragraph on significance for more details).

Molchanov et al. (2018): the scheme of their proof of the Theorem 1 of DSIVI is not directly applicable to IWHVI. Also, they require the bound to be averaged over $q(z)$, whereas the IWHVI gives a valid upper bound for any fixed $z$.

We are very thankful for the alternative proof of the Lemma C.2. While the existing Lemma is valuable as it provides an insight into the underlying generative process (Self-Normalized Importance Sampling), we will add the suggested proof for a reader's convenience.

To **Reviewer 4**: we introduced the term log marginal density to avoid confusion with the log marginal likelihood which is usually assumed to be $\log p(x)$ – the model's likelihood of the observed data, even though both terms mean logarithm of a marginal distribution of some joint distribution $\log \int p(\alpha, \beta|\gamma)d\beta$. Although the IWHVI bound could indeed be used to give an upper bound on the log marginal likelihood $\log p(x)$, it would require intractable and impractical posterior sampling. The paragraph starting on line 205 indeed meant the upper bound on the log marginal density, and we are thankful for pointing out the typo.

Missing related work: we will add citations, but they are outside of the scope of the paper: the Nonparametric VI only works for finite mixtures, and Likelihood-Free VI uses GAN-like estimation and loses lower bound guarantees in case of a suboptimal discriminator.

$K$ schedule: we observed it was beneficial to use as large $K$ as is affordable, however large number of samples seemed to cause computational instabilities in the early stages of training. 90% of training is done with $K = 50$.

UIVI: we omitted it because it did not scale to our VAE experiment setup. From the algorithmic point of view, MCMC-based methods are inherently sequential and thus are not amenable to parallelization. In particular, UIVI uses 5 samples from HMC with 5 leap-frog steps, and 5 burn-in steps, which results in 5*10 backward and 5*10 + 10 forward (including MH correction) passes done sequentially. Even if one runs 5 independent chains to obtain 5 samples in parallel, the burn-in is still required to decorrelate $\varepsilon$ and $\varepsilon'$, limiting the potential speed-up to 2x. In contrast, IWHVI/SIVI are much more parallel, as samples are independent and can be processed simultaneously. Originally UIVI was benchmarked on CPU in which case all methods indeed perform comparably time-wise (UIVI: 0.18sec/iter, IWHVI K=100: 0.14s/it, SIVI K=100: 0.13s/it). However, in our implementation in TensorFlow (with TensorFlow Probability for MCMC) for GPU we observed that IWHVI (0.02s/it) / SIVI (0.02s/it) are almost 8-9 times faster than UIVI (0.13s/it), and 7 times faster if one runs 5 parallel chains (0.11 s/it). We can add such experiments by the camera-ready deadline though.

Run times: are listed in the previous paragraph and are in absolute agreement with our discussion of complexity in the sec. 4.1, which predicts the actual SIVI/IWHVI run times to be indistinguishable.

Finally, we would like to address the **significance of the work**. We believe that our method significantly improves upon SIVI for several reasons. ①: it provides an upper-bound analog of the IWAE lower bound, whereas SIVI only gives an upper-bound analog of the IWAE lower bound for a special case of *the proposal distribution $q(z|x)$ fixed to be the prior $p(z)$*. This is not how IWAE bounds are typically used. ②: using the prior as a proposal forces the generative model of $x$ to adapt to such choice, that is, the true posterior $p(z|x)$ has to be close to the prior $p(z)$ (because of the gap), which negatively affects the mutual information (MI) between $x$ and $z$ and thus diminishes effectiveness of latent variable modeling (this is because each $z$ carries little information about $x$). Similar reasoning holds for SIVI/IWHVI, where we consider a (conditional) generative model with joint $q(z, \psi|x)$, true intractable "posterior" $q(\psi|x, z)$ and the "prior" $q(\psi|x)$. We validate this intuition with empirical results in fig. 1b: IWHVI with 5 samples has MI between $\psi$ and $z$ two times higher than SIVI with 50 samples. ③: we also note that despite IWHVI's similarity to IWAE lower bound and relations to SIVI/HVM, it is not obvious how to obtain the IWHVI bound in the absence of our theorem. We also render HVM and SIVI as special cases of a more general technique.

[Meta-Review · NeurIPS 2019]

This paper proposes a new tighter lower bound for variational inference by using auxiliary variable methods. R1 and R3 found the work significant and interesting. R4 had some concerns, some of which were resolved by authors' response. R4 decided to increase their score, however they do still have some concerns. I recommend the authors to take R4's comments into account and either remove the erroneous statements mentioned by them, or add explanation/proofs to show that the statements are correct, or at the very least clarify them so that such confusion does not arise to the reader in the future. There are many other small comments by the reviewers that need to be taken into account. In addition, the authors should do all the additional things they promised in the rebuttal. Given that this is done, this paper can be accepted.